# Histologic and Immunohistochemical Patterns in Lymphomatoid Papulosis: A Systematic Review of Published Cases

**DOI:** 10.3390/dermatopathology12010006

**Published:** 2025-02-12

**Authors:** Torben Fricke, Werner Kempf, Michael P. Schön, Christina Mitteldorf

**Affiliations:** 1Department of Dermatology, Venereology and Allergology, University Medical Center Göttingen, 37075 Göttingen, Germany; torben.fricke@stud.uni-goettingen.de (T.F.); michael.schoen@med.uni-goettingen.de (M.P.S.); 2Kempf und Pfaltz Histologische Diagnostik, 8050 Zürich, Switzerland; werner.kempf@uzh.ch; 3Department of Dermatology, University Hospital Zürich, 8091 Zürich, Switzerland

**Keywords:** lymphomatoid papulosis, CD30+ lymphoproliferative disorders, subtypes, histopathology, immunohistochemistry

## Abstract

Based on histologic and genetic patterns, the current World Health Organization (WHO) classification distinguishes six subtypes of lymphomatoid papulosis (Lyp). The aim of our article was to analyze the frequency of histologic and immunohistochemical features of different Lyp subtypes reported in the literature. We used PubMed advanced search builder to systematically review and evaluate English and German literature of Lyp from 1968 to April 2022. We considered only papers in which histopathologic features were mentioned in detail. We identified 48 publications with a total of 518 cases. The diagnoses were based on the diagnostic criteria at the time of publication. In Lyp A and Lyp B a CD8+ phenotype was more often reported than expected (53% and 52%, respectively). A double positive phenotype (CD4+/CD8+) was found in 28% of Lyp E and a double negative (CD4-/CD8-) in 50% of Lyp with 6p25.3 rearrangement. High rates of folliculo- and syringotropism were reported in both Lyp A and B. Surprisingly, strong epidermotropism occurred in 20/38 (53%) cases reported as Lyp B and in 43/64 (67%) of Lyp D cases. The predominating phenotype in Lyp D was CD8+, while TIA-1/granzymeB/perforin expression was reported in 37/46 (80%), and CD56 was expressed in 13/47 (28%) of the investigated cases. The limitation of the data is due to the retrospective approach with diagnostic criteria changing over time and on a case selection in some publications. However, the data indicate that the Lyp subtypes overlap more than assumed. They also show that a prospective study is needed to obtain valid data on the frequency distribution of certain histopathologic criteria.

## 1. Introduction

The group of CD30+ lymphoproliferative disorders (LPDs) includes lymphomatoid papulosis (LyP) and primary cutaneous anaplastic large cell lymphoma (pcALCL) [1,2,3,4,5,6]. LyP is characterized by recurrent, self-healing (waxing and waning) papules or small nodules with an indolent course, whereas pcALCL typically presents as solitary or grouped tumors, often with ulceration. A multifocal occurrence is observed in only 20% of cases. Histologically, LyP type C and pcALCL are indistinguishable, both being characterized by cohesive sheets of large CD30+ tumor cells with a CD30 expression exceeding 75%. CD30+ LPDs represent 25% of all cutaneous T cell lymphomas (CTCLs), thereby being the second most common CTCL entity. Lyp itself accounts for around 12% [2,3,7].

Clinically, Lyp is characterized by waxing and waning papules and nodules [5,8,9,10,11]. Lyp occurs mostly in adults but may also affect children and adolescents [11,12,13,14,15,16,17]. The lesions are mainly found on the trunk and extremities, but they can also appear in the sacral and perianal areas, as well as in atypical regions like the face [11,15,18,19,20].

In contrast to the rather uniform clinical presentation, the histopathologic and genetic heterogeneity of Lyp is the basis for the current distinction of a total of six subtypes by the World Health Organization (WHO) [21,22,23,24]. The classification of Lyp into subtypes was originally developed in order to have certain histological differential diagnoses in mind and to avoid misdiagnosis.

Lyp A is the most common form accounting for approximately 80% of all cases. It is characterized by a dense dermal infiltrate composed of large, atypical lymphoid CD30+ cells arranged as single cells or in aggregates and accompanied by numerous inflammatory cells, especially small reactive lymphocytes, neutrophils, and eosinophils [18,21,25,26].

Lyp B is rare and accounts for less than 5% of all Lyp cases. It shows rather small atypical lymphoid cells with cerebriform nuclei that express CD30 and predominantly CD4 [1,21,26,27]. These cells infiltrate the basal and suprabasal layers of the epidermis. The occurrence of CD30 negative tumor cells is also not unusual [21,23,24].

Lyp C is the second most common subtype of Lyp. It features a marked dermal infiltrate of large CD30+ lymphoid cells arranged in cohesive sheets and co-expressing CD4 [1,21,25,28,29].

A rare subtype of Lyp, with <5% of all cases, is subtype D, which is characterized by a cytotoxic CD8+ phenotype. The small to medium-sized atypical tumor cells show a strong epidermotropism, sometimes pagetoid. The dermal infiltrate may display a deep dermal component with occasional infiltration of the subcutaneous tissue [1,21,30,31,32].

Subtype E clinically displays large necrotic “eschar”-like lesions reflecting microscopically angioinvasive and angiodestructive infiltrates. The CD30+ lymphocytes frequently co-express CD8 and occasionally CD56 [1,9,21,33,34].

The recently described Lyp with 6p25.3 rearrangement, defined by a chromosomal rearrangement of the DUSP22-IRF4 locus at 6p25.3, is characterized by a biphasic pattern with extensive epidermotropism of weakly CD30+ small to medium-sized lymphocytes and a dense dermal infiltrate comprising strongly CD30+ medium to large sized cells. It predominantly shows a double negative phenotype [1,21,35].

Further patterns were described but not (yet) addressed as separate subtypes by the WHO classification [1,3,24,36,37,38,39]. These patterns show an invasion of adnexal structures (follicular or syringotropic/eccrinotropic) and are sometimes accompanied by a granulomatous component [36,37,38,39,40,41,42,43,44,45,46,47,48].

The aim of our study was to critically re-evaluate the histopathologic criteria for the different Lyp subtypes with a special focus on overlapping histopathologic patterns. Cases with γ/δ T-cells were only considered if they could be assigned to one of the defined subtypes [49,50,51].

## 2. Materials and Methods

We systematically reviewed PubMed for all reports of Lyp from 1968 to April 2022 considering the histopathologic features of the different subtypes. We used the PubMed Advanced Search Builder to look up all titles and abstracts for the keywords “lymphomatoid papulosis”, “type”, and “histopathology”. Additionally, we searched for further publications in the references of the articles found. Original studies but also case reports were included, if histopathologic features were investigated or mentioned in detail. Only articles in English and German were considered. Basic clinical parameters (sex and age) were also recorded.

Only studies that provided individual information on the histologic features of the cases presented were included. The individual cases were examined according to the presence or absence of the histological features and then documented in tabular form. The extracted data then was collected in several different tables. Mean values and thus the probability of occurrence for the respective histological feature were calculated for the individual subtype.

These studies were reviewed independently by the main author and the information was extracted without any kind of automation tools. The data were analyzed and extracted independently and without the influence of other persons or institutions. The review has not been registered, and there is no review protocol available. All data used and analyzed in this review are publicly available in the databases mentioned above.

All authors state that this review followed the PRISMA Guidelines and address all elements of the PRISMA checklist.

## 3. Results

### 3.1. Frequencies of Reported Subtypes

Forty-eight articles with 518 cases were included in this study. In the first place we originally screened 988 reports of which the majority were excluded for various reasons which can be seen in Figure 1. In the literature, Lyp A was the most frequently reported subtype (265/518 cases; 51%). The second most common subtypes were D (74/518; 14%) and B (73/518; 14%), respectively. Only 53/518 (10%) cases were subtype C, followed by subtype E 41/518 (8%) and 6p25.3 rearrangement subtype 12/518 (3%).

### 3.2. Basic Clinical Parameters

The mean age of the patients in the different subtypes, except for those with 6p25.3 rearrangement, ranged from 38 to 50 years. Occurrence in childhood was also reported. In contrast, in patients with Lyp with 6p25.3 rearrangement the mean age was 71 years (range: 68–88 years). So far, this subtype has never been reported in children, adolescents, or young adults.

Regarding gender, the male to female ratio was 4:1 in subtype B, 3:1 in subtype 6p25.3 rearrangement, 2:1 in subtype E, and 2:1 in subtype A. In contrast, the male to female ratio was nearly equal in subtypes C and D.

### 3.3. Histologic Parameters

All investigated histologic patterns, the immunophenotype, and genetic features which could be extracted from the articles, are, respectively, shown in the Appendix A.

#### 3.3.1. Lyp Subtype A

For subtype A, we identified 13 articles with 265 cases containing information about the histopathologic features [25,26,28,48,52,53,54,55,56,57,58,59,60]. The extent of the examination of certain aspects widely differs amongst various publications. Details about the immunophenotype were only given in 130 cases, with a highly variable extent of investigated markers. A total of 45 cases from Willemze et al. [60] were excluded due to missing detailed individual information for each case.

A CD8+ phenotype (69/130 = 53%) was more commonly reported than a CD4+ one (54/130 = 42%). Two of 130 cases presented a double-positive (CD4+/CD8+) and 5 of 130 a double-negative (CD4-/CD8-) phenotype. Neutrophils and eosinophils were not found in 41% (88/216) and 39% (85/217) of the cases, respectively [1,59]. In 26% (51/197) folliculotropism and in 27% (56/209) syringotropism was present [25,26,52].

The main features of Lyp subtype A are summarized in Figure 2. The frequency of all investigated features is shown in Appendix A.

#### 3.3.2. Lyp Subtype B

We found 11 publications with 73 cases that addressed the histopathologic features of subtype B [25,26,27,28,52,56,59,60,61,62,63]. There were 19 cases from Willemze et al. [60] excluded due to missing detailed individual information for each case. Fourteen percent of Lyp subtype B were CD30 negative [25,28,52,56], which is in line with the existing definition [21]. The immunophenotype was only addressed in 46 cases [25,26,27,28,52,56,59,63]. A total of 52% (24/46) of these cases were CD8+, while 43% (20/46) exhibited a CD4+ phenotype [26,52,63]. One case each was double negative or double positive, respectively [25,59]. In 53% (20/38) a pagetoid epidermotropism was described, while basal epidermotropism was found in 100% [25,26,27,28,52,57,59,62]. Folliculotropism (15/33 = 46%) and syringotropism (14/36 = 39%) were common.

The main features of Lyp subtype B are summarized in Figure 3. The frequency of all investigated features is shown in Appendix A.

#### 3.3.3. Lyp Subtype C

We found 10 publications for subtype C with 53 cases [25,26,27,28,29,59,64,65,66,67]. The immunophenotype was predominantly CD4+ (15/29; 52%), whereas 11/29 cases (38%) expressed CD8. It is remarkable that 57% (30/53) showed epidermotropism into the basal layers, and pagetoid epidermotropism was found in 13% (7/53).

Cytotoxic markers (TIA-1, perforin or granzyme B) were expressed in 79%, but only 19 cases were evaluated. Cytotoxic marker expression was not restricted to cases with a CD8+ phenotype [25,27,59,67].

The main features of Lyp subtype C are summarized in Figure 4. The frequency of all investigated features is shown in Appendix A.

#### 3.3.4. Lyp Subtype D

We identified 19 publications with overall 74 cases [27,28,30,31,32,59,68,69,70,71,72,73,74,75,76,77,78,79,80,81]. Although subtype D is considered rare, this form, together with subtype B, was the second most frequently published. As expected, almost all cases were CD8+ (64/68 = 94%). Double positive and double negative phenotypes were present in 3/68 (4%) and 1/68 (2%), respectively. Another important criterion for subtype D is a pagetoid epidermotropism, which was, however, found in only 67% (43/64) of the reported cases. An epidermotropism into the basal layer was observed in 57/67 (85%) cases. One case without any epidermotropism was diagnosed as subtype D [32].

Cytotoxic markers (TIA-1, granzyme or perforin) were identified in 80% (37/46) of the investigated cases. CD56 was examined in 47 cases and was positive only in 28% (13/47) cases. Remarkably, the γ/δ-phenotype was expressed in 2 of 4 evaluated cases (50%). The α/β-phenotype was displayed in 10/12 (83%) cases.

The main features of Lyp subtype D are summarized in Figure 5. The frequency of all investigated features is shown in Appendix A.

#### 3.3.5. Lyp Subtype E

Twelve publications with 41 cases could be examined [9,33,34,80,82,83,84,85,86,87,88,89]. A double positive phenotype was common (10/36 cases; 28%). Sixteen of 36 (44%) cases showed a CD8+ phenotype, whereas 9 of 36 (25%) were positive for CD4 [9,81,84,87]. One case was double negative [90].

The characteristic feature of angiocentric infiltration was found in 83% (34/41), while 17% (7/41) of the cases showed only a perivascular arrangement [80]. A vascular occlusion was present in all cases and hemorrhage was reported in 25 of 30 (83%) cases. An extensive epidermal necrosis was described in 18 of 24 (75%) [9,34,89]. An admixture of neutrophils was described in 30 of 36 (83%) cases. Eosinophils were also present in 22 of 35 (63%) cases.

The main features of subtype E are summarized in Figure 6. The frequency of all investigated features is shown in Appendix A.

#### 3.3.6. Lyp Subtype with 6p25.3 Rearrangement

Only 2 publications with a total of 12 cases have been published [35,90]. Half (6/12) of the cases displayed a double negative phenotype. Four cases (33%) were CD8+, one case (8%) was CD4+, and one case was CD4/CD8 double positive. All cases showed the typical biphasic pattern with dermal and pagetoid infiltrates.

The main features of Lyp subtype with 6p25.3 rearrangement are summarized in Figure 7. The frequency of all investigated features is shown in Appendix A.

## 4. Discussion

As in all retrospective data analyses, the results merely reflect the frequency distribution of Lyp subtypes, as well as the histopathologic and immunophenotypic features of reported cases in the literature. We only considered papers in which histopathologic features were mentioned and we also included case reports. Many studies were published before all currently known Lyp subtypes had been described. Hence, some cases from older publications would probably be assigned differently today. Moreover, there is a tendency that rare and uncommon observations were more commonly published and some authors did a preselection of their study groups.

The WHO classification [21,23,24] mentioned a mean age for patients with Lyp of 45 years. It also states that Lyp can also occur in children. Our study showed that mean ages reported in Lyp subtypes A–E ranged from 38–50 years. Lyp subtypes C and B have the lowest average age at 38 years, while the average age of Lyp subtype E was 50 years. The average age of subtype 6p25.3 rearrangement is significantly higher at 71 years with a minimum age of 67 years [35,90]. The latter subtype has never been reported in children, adolescents, or young adults [35,90].

As mentioned by the WHO [21,24], subtypes A, B, and E and 6p25.3 rearrangement showed a male to female ratio between 2:1 and 4:1. In contrast, the gender distribution in subtypes C and D was nearly equal.

When examining the individual subtypes, we also encountered some differences or uncommon histopathologic and immunophenotypic features compared to the previously established definitions.

The more commonly reported CD8 phenotype in Lyp subtype A (53%) and Lyp subtype B (52%) might be related to a selection bias [25,26,27,28,52,55,56,57,58,59,60,61,62,63]. As a limitation, not all the papers stated whether this phenotype is restricted to the CD30-positive cells or if the surrounding infiltrate of small lymphocytes was also considered. Lyp subtype A is furthermore characterized by “[…] intermingled with numerous inflammatory cells, including small lymphocytes, neutrophils and/or eosinophils” [21]. Indeed, admixed small lymphocytes were found in nearly all cases, but in about 40% of the cases no neutrophils or eosinophils were present [25,26,48,52,54,57,58,59]. That shows that originally described type-defining criteria are not reproducible in the literature. In the daily routine it might be difficult to decide if a case without neutrophils or eosinophils fits with Lyp subgroup A and, more important, if the mentioned main differential diagnoses are correct and complete. In addition, it may be debated whether and to what extent the high proportion of folliculotropic and syringotropic cases justifies classification as a separate subtype. It is however important to consider that only two of 13 studies investigated folliculotropism and syringotropism in Lyp subtype A and that the reported frequencies varied significantly, e.g., for folliculotropism, between 5% and 49% [25,26]. On the one hand, this demonstrates that redefining a new folliculotropic subtype would create more overlap; on the other hand, by not considering this distinct feature, the important differential diagnosis of folliculotropic mycosis fungoides would remain unaddressed.

In Lyp subtype B, one study had an unusual high number of extensive epidermotropism in combination with a high number of CD8 expressing cases [26]. This raises the question of whether those cases rather represent Lyp subtype D, first mentioned in 2006 [70] and finally described as a new variant in 2010 [30]. Fourteen percent of Lyp subtype B were CD30 negative, which is in line with the existing definition [21]. As in Lyp subtype A, we also observed high rates of folliculotropic and syringotropic infiltration, but these were also only mentioned in two reports [26,52]. In 9 of 11 studies these features were not investigated. These adnexotropic patterns also occurred in other Lyp subtypes but were only rarely investigated in the subtypes C–E and Lyp subtype with 6p25.3 rearrangement.

Most (52%) of the Lyp subtype C cases showed the expected CD4 phenotype [25,26,29,67]. A predominance for CD8, however, was present in 38% of the cases. Eleven of 29 biopsies (38%) showed a CD4-CD8+ phenotype. Seven of these 11 cases were from a study that had only included CD8+ cases and 2 of 11 cases came from a study that had selected only gamma/delta+ cases (two of which were also CD8+) [59,77]. The presence of cohesive sheets was only mentioned explicitly in four cases [28,29,65,67]. A reason for this unusually low number might be that cohesive sheets are a defining criterion for Lyp subtype C and therefore a prerequisite for this subtype. Notably, Lyp subtype C also has a high proportion of basal epidermotropism (57%) and strong (sometimes pagetoid) epidermotropism in 13% of cases [25,26,27,28,64,65,66,67].

Subtype D is a well-studied type with 74 biopsies evaluable in this analysis. Including the first description by Saggini et al. [30], a total of 19 publications with detailed histopathologic examinations have been published so far [27,28,30,31,32,59,69,70,71,72,73,74,75,76,77,78,79,80,81]. Extensive and sometimes pagetoid epidermotropism, which was described as a key feature, remarkably was documented only in 67% and, even more surprisingly, reported to be absent in 33% of the cases [32,78,80]. This further illustrates that type-defining criteria have not been consistently applied in the literature. Since the CD8+ phenotype is part of the definition, 94% of the cases were CD8+. Additionally, two cases each were CD4/CD8 double positive or double negative, one of which was part of the first description [30]. In the works of Bertolotti et al. and Bergqvist et al. [71,80] there are a total of 3 cases with a CD4+/CD8+ phenotype. CD56 was positive in 13 of 47 (28%) investigated cases [30,59,71,72,77,80,81]. The cytotoxic phenotype could be identified in 80% of the investigated samples [27,30,31,69,70,71,72,73,76,77,78,80,81].

Lyp subtype E is defined by an angiocentric and angioinvasive growth pattern [9,21]. This criterion, however, was not met in 8% of the cases [80]. Interestingly, 7 cases presented only a perivascular arrangement of the infiltrates [80]. One has to be aware that most of the other Lyp subtypes also showed a high frequency of a perivascular distribution. Vascular occlusion was found in all cases of Lyp subtype E, but was not frequently reported in other Lyp subtypes. CD8 was in fact the most common immunophenotype (44%), while 28% of the cases were CD4/CD8 double positive and 25% were positive for CD4. This demonstrates that the immunophenotype is broader than initially described and in line with our personal experience.

The Lyp subtype with 6p25.3 rearrangement is extremely rare, with only 12 published cases [35,90]. Since the first description of the Lyp subtype with 6p25.3 rearrangement in 2013 [35], only one additional case was reported [90]. This may in fact reflect the actual rarity of this subtype, but also the rarely performed molecular genetic analysis of the DUSP22/IRF4 locus, which was only investigated in 25 cases in the literature [35,80,90]. The reported characteristics are essentially based on a case series, which can lead to quite homogeneous features. The most common phenotype was CD4 and CD8 double negative, accounting for 50% of the cases. One third of the cases was CD8 positive. In contrast to most other subtypes, eosinophils were only rarely present in less than 20% of Lyp subtype with 6p25.3 rearrangement [35].

## 5. Conclusions

This retrospective analysis highlights the challenges and limitations in the categorizing of Lyp subtypes, as many studies relied on older definitions and incomplete reporting of histopathologic features. The different subtypes of Lyp do not have a different prognosis, however, the subtyping was primarily intended to memorize histological differential diagnoses and to avoid misdiagnoses. Our results demonstrate variability in the frequency of specific immunophenotypic and histopathologic characteristics, suggesting that current subtype definitions may not be universally applicable. Several subtypes exhibit overlapping features or additional distinct patterns, like folliculotropism and syringotropism, which were not considered when discussing histologic differential diagnoses. Therefore, future prospective studies that incorporate both established and novel histological criteria are essential for refining the classification system and improving the accuracy of diagnoses.

## Figures and Tables

**Figure 1 dermatopathology-12-00006-f001:**
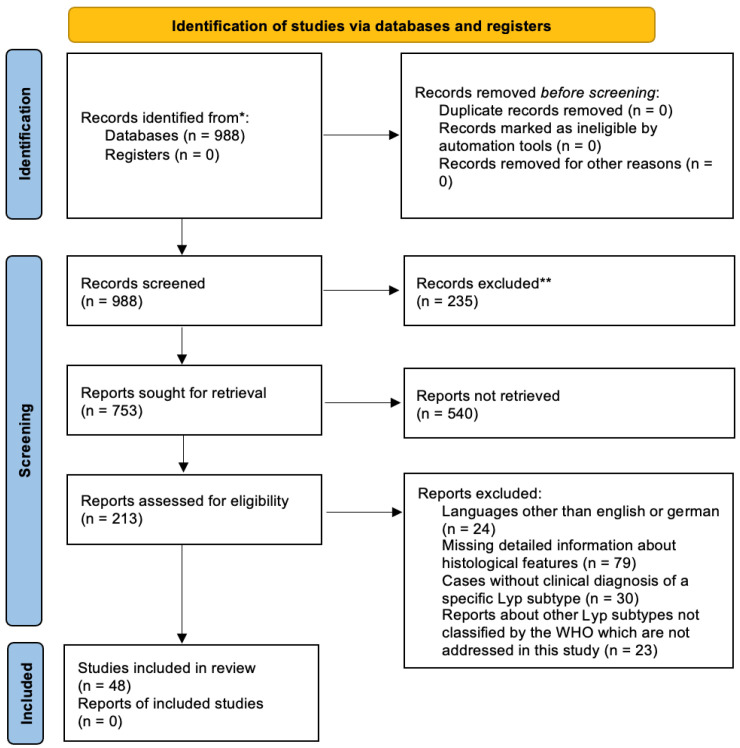
PRISMA flow diagram. (* Consider, if feasible to do so, reporting the number of records identified from each database or register searched (rather than the total number across all databases/registers). ** If automation tools were used, indicate how many records were excluded by a human and how many were excluded by automation tools).

**Figure 2 dermatopathology-12-00006-f002:**
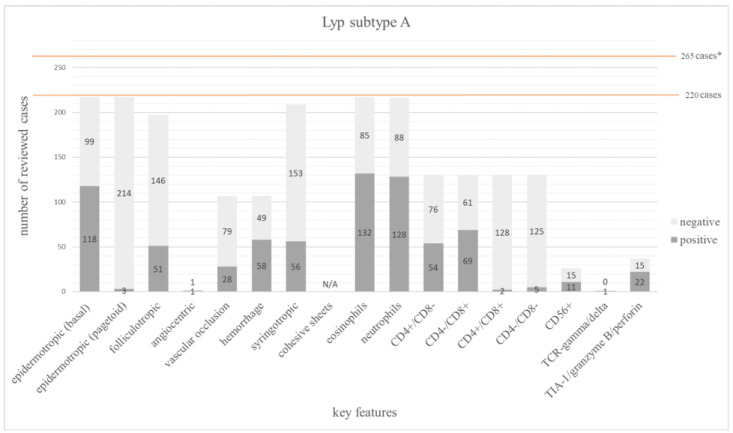
The bar chart depicts 17 main histopathologic and immunophenotypic features evaluated in 220 cases of Lyp subtype A. The dark grey columns show the number of positive cases for each feature, the light grey columns show the negative ones. The orange line shows the total number of all investigated cases. * 45 cases from Willemze et al. [60] were excluded due to missing detailed individual information for each case.

**Figure 3 dermatopathology-12-00006-f003:**
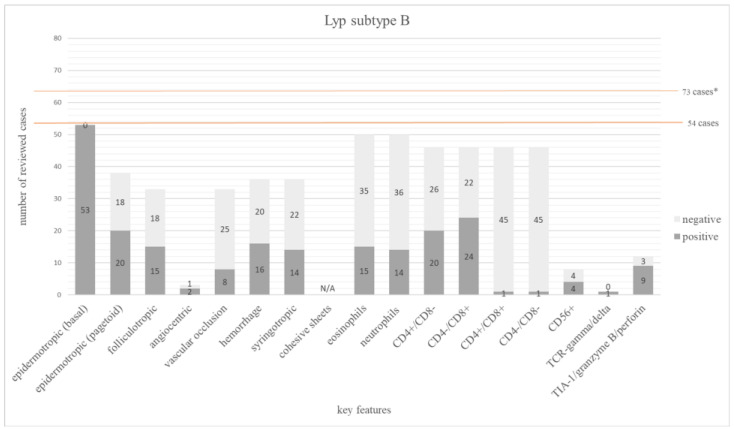
The bar chart depicts 17 main histopathologic and immunophenotypic features evaluated in 54 cases of Lyp subtype B. The dark grey columns show the number of positive cases for each feature, the light grey columns show the negative ones. The orange line shows the total number of all investigated cases. * 19 cases from Willemze et al. [60] were excluded due to missing detailed individual information for each case.

**Figure 4 dermatopathology-12-00006-f004:**
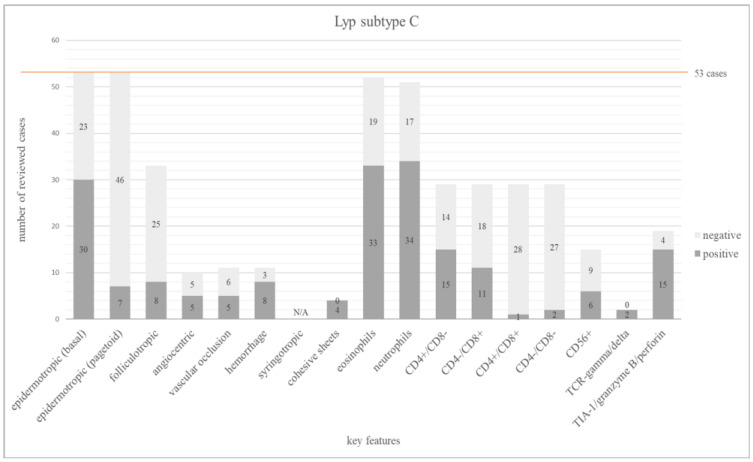
The bar chart depicts 17 main histopathologic and immunophenotypic features evaluated in 53 cases of Lyp subtype C. The dark grey columns show the number of positive cases for each feature, the light grey columns show the negative ones. The orange line shows the total number of all investigated cases.

**Figure 5 dermatopathology-12-00006-f005:**
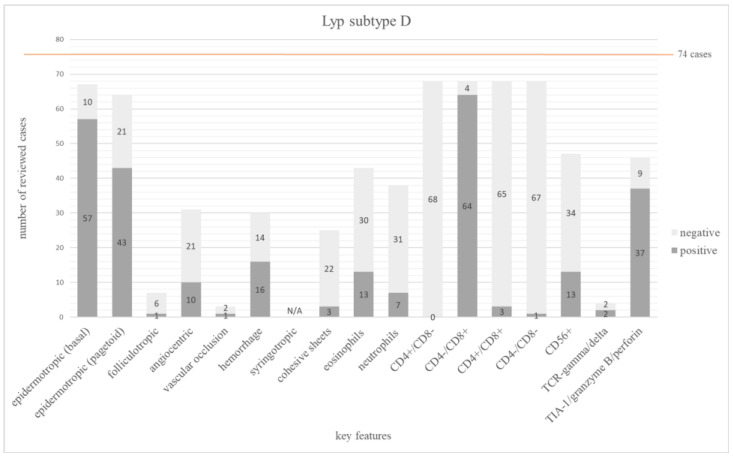
The bar chart depicts 17 main histopathologic and immunophenotypic features evaluated in 74 cases of Lyp subtype D. The dark grey columns show the number of positive cases for each feature, the light grey columns show the negative ones. The orange line shows the total number of all investigated cases.

**Figure 6 dermatopathology-12-00006-f006:**
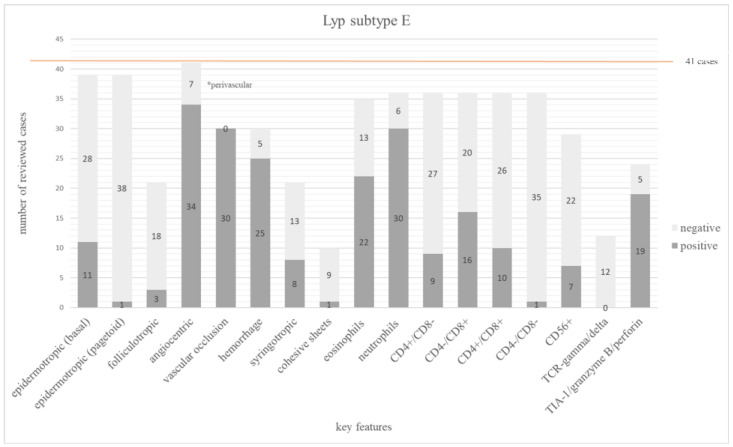
The bar chart depicts 17 main histopathologic and immunophenotypic features evaluated in 41 cases of Lyp subtype E. The dark grey columns show the number of positive cases for each feature, the light grey columns show the negative ones. The orange line shows the total number of all investigated cases.

**Figure 7 dermatopathology-12-00006-f007:**
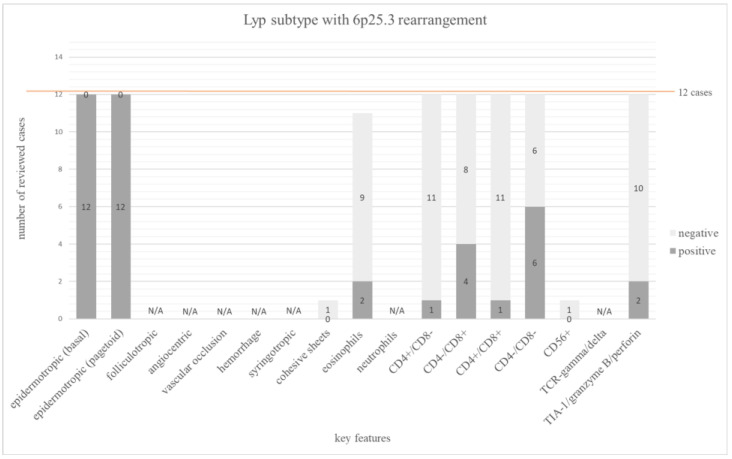
The bar chart depicts 17 main histopathologic and immunophenotypic features evaluated in 12 cases of Lyp subtype with 6p25.3 rearrangement. The dark grey columns show the number of positive cases for each feature, the light grey columns show the negative ones. The orange line shows the total number of all investigated cases.

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
