# Peer review of "Histologic and Immunohistochemical Patterns in Lymphomatoid Papulosis: A Systematic Review of Published Cases"

_dermatopathology, 2025, doi:10.3390/dermatopathology12010006_

Round 1

Reviewer 1 Report

Comments and Suggestions for Authors

In this laborious paper, the authors searched for publications on lymphomatoid papulosis (Lyp) over the past 50 years and retrospectively analyzed the Lyp that had been reported up to date. They found that several subtypes defined by the WHO have overlapping histopathological and immunohistochemical features. This paper questions the validity of the WHO classification and is considered an important paper.

Comments.

1.    Are there differences in prognosis between the six subtypes? If so, please mention them in the text. If there aren't, what do the authors think is the significance of the classification into six categories?

2.    It would be good to explain the differences between Lyp and primary cutaneous anaplastic large cell lymphoma (pcALCL), which is one of the CD30-positive lymphoproliferative disorders (LPD), in the introduction or discussion.

3.    The way the data are presented is too detailed, making it difficult to grasp the overall picture. I think a table summarizing the characteristics of each subtype would help understanding.

Author Response

Comment 1:

In this laborious paper, the authors searched for publications on lymphomatoid papulosis (Lyp) over the past 50 years and retrospectively analyzed the Lyp that had been reported up to date. They found that several subtypes defined by the WHO have overlapping histopathological and immunohistochemical features. This paper questions the validity of the WHO classification and is considered an important paper.

Response 1:

Thank you very much for your comment.

Comment 2:

Are there differences in prognosis between the six subtypes? If so, please mention them in the text. If there aren't, what do the authors think is the significance of the classification into six categories?

Response 2: 

Thank you very much for raising up this important point. The different Lyp subtypes do not have a different prognosis. The idea of classifying in six subtypes is to mention histopathological differential diagnosis. However, our paper shows that on the one hand the restriction of subtypes only covers some histopathologic features in Lyp and on the other hand the overlap does often not allow a clear assignment to one subtype in most cases. In our opinion, the subtyping is less helpful than intended. We added a paragraph to clarify the significance of the classification in the introduction section on page 2, lines 51-52.

Comment 3:

It would be good to explain the differences between Lyp and primary cutaneous anaplastic large cell lymphoma (pcALCL), which is one of the CD30-positive lymphoproliferative disorders (LPD), in the introduction or discussion.

Response 3:

Thank you for the important comment. We have added a paragraph explaining the clinical and histological differences between Lyp and pcALCL in the introduction on page 1, lines 34-40.

Comment 4:

The way the data are presented is too detailed, making it difficult to grasp the overall picture. I think a table summarizing the characteristics of each subtype would help understanding.

Response 4:

Thank you for your comment. To clarify and getting the overall picture we highlighted the key features of the figure in each of the corresponding suppl. tables. We additionally clarified the interpretation of the results in the discussion to get a better overall picture. We have also completely revised the conclusion and emphasized the key points more strongly.

Reviewer 2 Report

Comments and Suggestions for Authors

These rare diseases with frequent overlap need to be studied with new diagnostic criteria

Author Response

Comments 1:

These rare diseases with frequent overlap need to be studied with new diagnostic criteria.

Response 1:

Thank you for your comment. We agree that a prospective analysis of Lyp should be performed according to currently valid and possibly new histological criteria. However, we first wanted to summarize and evaluate the existing data in the literature in order to show that a prospective analysis is necessary and to redefine the criteria to be examined here. This has now been done and the prospective data are currently being analyzed. We have inserted a corresponding note on page 9, lines 309-310.

Reviewer 3 Report

Comments and Suggestions for Authors

The authors conducted a systematic review of published literatures concerning lymphomatoid papulosis (Lyp) cases to re-evaluate the histopathologic criteria for the different Lyp subtypes. This reviewer has read the manuscript with great interest and has some concerns.

1. Tumor cells in Lyp B can be negative for CD30 as described in the Discussion section. That should also be added in the Introduction section, the Results section, “Lyp subtype B”, and Figure 2.

2. The authors conducted this study to “critically re-evaluate the histopathologic criteria for the different Lyp subtypes”, but the Discussion section seems to be little more than a repetition of the Results section. Based on the results, the difference between type B and type D might not be clearly identified by many clinicians. Perhaps these subtypes could be unified. Anyway, the Discussion should be more deepened based on the authors’ opinion but not on the opinion of this reviewer.

3. The titles of Supplementary Tables are missing.

Author Response

Comment 1:

The authors conducted a systematic review of published literatures concerning lymphomatoid papulosis (Lyp) cases to re-evaluate the histopathologic criteria for the different Lyp subtypes. This reviewer has read the manuscript with great interest and has some concerns.

Response 1:

Thank you very much for evaluating our manuscript and for the suggestions and comments.

Comment 2:

Tumor cells in Lyp B can be negative for CD30 as described in the Discussion section. That should also be added in the Introduction section, the Results section, “Lyp subtype B”, and Figure 2.

Response 2:

Thank you for pointing out this very important information. To emphasize this point we added a sentence to the introduction on page 2, line 51-52. We also adapted the results part. We stated the articles where CD30-negative cases occurred in the results section as recommended on page 4, line 128-129. However, in the figures we focused on a few key features, which were identical in all Lyp subtypes.

Comment 3:

The authors conducted this study to “critically re-evaluate the histopathologic criteria for the different Lyp subtypes”, but the Discussion section seems to be little more than a repetition of the Results section. Based on the results, the difference between type B and type D might not be clearly identified by many clinicians. Perhaps these subtypes could be unified. Anyway, the Discussion should be more deepened based on the authors’ opinion but not on the opinion of this reviewer.

Response 3:

Thank you for your comment. The point does indeed show the problem that we noticed during the analysis. The results clearly show that it is often not possible to clearly categorise the subtypes. The criteria are not always clearly defined in the literature. This leads to cases being categorised as type B, for example, which today would correspond to subtype D. We also think that this type of subtyping can be dispensed with, but the broad histological spectrum of Lyp should continue to be pointed out in order to avoid histological misjudgements. We have added our personal assessment and critical view of the current subtyping to the discussion and the conclusion.

Comment 4:

The titles of Supplementary Tables are missing.

Response 4:

Thank you for your comment. We added a title for each table.

Round 2

Reviewer 3 Report

Comments and Suggestions for Authors

The authors revised the manuscript well.